# An Improved Lightweight User Authentication Scheme for the Internet of Medical Things

**DOI:** 10.3390/s23031122

**Published:** 2023-01-18

**Authors:** Keunok Kim, Jihyeon Ryu, Youngsook Lee, Dongho Won

**Affiliations:** 1Department of Electrical and Computer Engineering, Sungkyunkwan University, 2066 Seobu-ro, Jangan-gu, Suwon-si 16419, Republic of Korea; 2Department of Computer Science and Engineering, Sungkyunkwan University, 2066 Seobu-ro, Jangan-gu, Suwon-si 16419, Republic of Korea; 3Department of IT Software Security, Howon University, 64 Impi-myeon, Howondae 3-gil, Gunsan-si 54058, Republic of Korea; 4Department of Computer Engineering, Sungkyunkwan University, 2066 Seobu-ro, Jangan-gu, Suwon-si 16419, Republic of Korea

**Keywords:** Internet of Medical Things (IoMT), user authentication, biometric, healthcare, healthcare IoT

## Abstract

The Internet of Medical Things (IoMT) is used in the medical ecosystem through medical IoT sensors, such as blood glucose, heart rate, temperature, and pulse sensors. To maintain a secure sensor network and a stable IoMT environment, it is important to protect the medical IoT sensors themselves and the patient medical data they collect from various security threats. Medical IoT sensors attached to the patient’s body must be protected from security threats, such as being controlled by unauthorized persons or transmitting erroneous medical data. In IoMT authentication, it is necessary to be sensitive to the following attack techniques. (1) The offline password guessing attack easily predicts a healthcare administrator’s password offline and allows for easy access to the healthcare worker’s account. (2) Privileged-insider attacks executed through impersonation are an easy way for an attacker to gain access to a healthcare administrator’s environment. Recently, previous research proposed a lightweight and anonymity preserving user authentication scheme for IoT-based healthcare. However, this scheme was vulnerable to offline password guessing, impersonation, and privileged insider attacks. These attacks expose not only the patients’ medical data such as blood pressure, pulse, and body temperature but also the patients’ registration number, phone number, and guardian. To overcome these weaknesses, in the present study we propose an improved lightweight user authentication scheme for the Internet of Medical Things (IoMT). In our scheme, the hash function and XOR operation are used for operation in low-spec healthcare IoT sensor. The automatic cryptographic protocol tool ProVerif confirmed the security of the proposed scheme. Finally, we show that the proposed scheme is more secure than other protocols and that it has 266.48% better performance than schemes that have been previously described in other studies.

## 1. Introduction

The Internet of Medical Things (IoMT) represents a combination of the healthcare field and the IoT ecosystem that can be used to create, collect, transmit, and analyze medical data through the connection of various healthcare IT systems, healthcare sensors, and healthcare management programs [1,2]. With the continued evolution of IoT technology and the outbreak of COVID-19, IoMT gaining increased interest as it can enable personalized medical information management, real–time health tracking and monitoring but also remote treatment [3].

However, there may be various security problems in the IoMT environment when dealing with sensitive medical information [4,5], such as the following:1.First, if malicious cyberattacks can take control of healthcare sensors attached to a patient’s body, this might not only result in inaccurate data collection but also put the patient’s health at risk.2.Second, malicious cyberattacks can expose sensitive patient data and medical information.3.Third, since IoMT uses low–power wearable healthcare sensors, the protocol used is not lightweight, and it therefore may be difficult to operate normally or to provide real–time service due to the need for time–consuming computation.

Therefore, further develop the IoMT environment, it is crucial to maintain security for medical systems and IoT devices and to support lightweight security protocols for implementing them.

We implemented and analyzed a suitable scheme for IoMT using the following method. For safety, we not only use a simple password base but also introduce a fuzzy extractor, i.e., it is a biometric-based authentication method. We also propose a system model and an attack model to complement the weaknesses of Masud et al. [6]. We analyze the vulnerabilities of Masud et al. [6] based on the system model and attack model and propose our new IoMT scheme. We analyze the safety of the proposed scheme using a formal analysis and an informal analysis and calculate the cost of computation. Lastly, we analyze how efficient the computational cost is compared to those of other schemes.

### 1.1. Our Contribution

We proposed a secure and lightweight user authentication scheme for IoMT by improving on Masud et al. [6]’s scheme by addressing the possible threats involved. In summary, we make the following contributions:1.First, to overcome offline password-guessing attacks, we added biometrics authentication methods that can only authenticate the user when an actual user is present. Further, to protect against replay attacks, we added logic to ensure that the gateway authenticates the user and the freshness of the user’s message in the authentication phase. Finally, to overcome privileged insider attacks, we deleted the secret information that shared between the user and the sensor in the registration phase immediately following the registration phase.2.We proposed a lightweight security protocol that mainly uses a hash function and XOR operation to run low-spec healthcare sensors.3.The proposed scheme is designed to protect against various security threats such as offline password guessing attacks, privileged insider attacks, user impersonation attacks, replay attacks, and session key disclosure attacks. It also ensures user anonymity.

### 1.2. Organization of Our Paper

The rest of this paper is organized as follows: Section 3 presents the preliminaries about the fuzzy extractor, system model, and attack model. Section 4 presents the scheme reported by Masud et al. [6]. Section 5 demonstrates the scheme reported by Masud et al. [6]. Section 6 presents our improved scheme. Section 7 provides formal and informal security analysis. Section 8 provides a performance analysis. Section 9 provides discussion of performance. Finally, Section 10 presents our conclusion.

## 2. Related Work

Even until recently, most healthcare systems could only be accessed using a password. Password-based authentication [7,8] is the most popular method for user authentication. However, it is unfortunately not suitable for use in a sensitive system that requires strong security because it contains various security threats. For example, password-based authentication can lead to unintentional password sharing when someone looks over the user’s shoulder and sees the user’s password. Anyone who knows this password can access the system on behalf of the user. Moreover, a password guessing attack is possible, in which the attacker guesses the user’s password and attempts authentication until finding success.

Password-based authentication schemes can also be exposed to various security threats, so to overcome the security threats of password-based authentication, two–factor authentication using a smart card has been introduced. In 2012, Wu et al. [9] proposed a secure authentication scheme for TMIS using a smart card and a password. However, Debiao et al. [10] pointed out that this scheme was vulnerable to impersonation attacks and insider attacks. Meanwhile, Wei et al. [11] pointed out that the scheme was vulnerable to offline password guessing attacks. To overcome these problems, Debiao et al. [10] proposed a more secure authentication scheme using a smart card and a password. However, Wei et al. [11] pointed out that this scheme also was vulnerable to offline password guessing attacks if the user were to lose his or her smart card. Consequently, a three-factor authentication using a password, a smart card, and biometrics, i.e., fingerprint and face recognition, has been introduced to achieve a higher level of security [12,13,14,15]. Wu et al. [12] proposed an improved and provably secure three-factor user authentication scheme for wireless sensor networks. However, Ryu et al. [13] pointed out that this scheme was vulnerable to user impersonation attacks and that it could not preserve the user’s anonymity could not be preserved. A summary of each scheme’s weaknesses is presented in Table 1.

Recently, Masud et al. [6] proposed a lightweight and anonymity preserving user authentication scheme for IoT–based healthcare. In the present study, we specifically focused on the use of lightweight protocols to support resource–constrained devices. However, we found various security threats, since this paper only provides user authentication using a password to remain lightweight. While it is important to keep in mind that supporting lightweight security protocols in an IoMT environment is one of the most important considerations, maintaining security is the most foundational requirement. One of the fatal security threats to which Masud et al. [6]’s scheme is vulnerable is an offline-password attack. If an attacker steals a valid authentication message in any authentication phase that is communicated via a public channel, the attacker can guess the user’s valid password while offline. Masud et al. [6]’s scheme is also vulnerable to replay attacks and privileged insider attacks. Through such attack, an attacker could log in by stealing the user’s password and disguising themselves as the user. An attacker could also enter information on behalf of the user or have access to the user’s information as the user.

Although Masud et al. [6]’s scheme exhibited better performance to maintain security for IoMT, it still has a security challenge. Therefore, we proposed an improved lightweight user authentication scheme for IoMT that improved upon Masud et al. [6]’s scheme and reduces reduced the computational cost compared to related studies.

## 3. Preliminaries

In this section, we introduce the fuzzy extractor, the system model which we made, and the attack model. The details are as follows:

### 3.1. Fuzzy Extractor

Biometric information is the best way to authenticate and verify users [16,17,18]. In 2004, Dodis et al. [19] proposed the Fuzzy Extractor to obtain a unique bit string extracted from the biometric template. A fuzzy extractor is a tuple $\left(M,m_{e},l,\tau ,\epsilon \right)$ that has two algorithms Gen and Rep, which are expressed as follows [12]:(1)$$\mathrm{Gen}\left(w_{i}\right)=\left(R_{i},P_{bi}\right)$$

The above result means that the fuzzy extractor can generate the secret string $R_{i}$ and $P_{bi}$. $w_{i}$ is a $U_{i}$’s original collected biometric data.
(2)$$Rep\left(w_{i}^{{}^{\prime }},P_{bi}\right)=R_{i}$$

The above result means that the fuzzy extractor can recover the secret string $R_{i}$ that is generated by the Gen algorithm.

### 3.2. System Model

Figure 1 shows the system model of the proposed scheme. In our scheme, there are three entities: the User, Medical Service Gateway (Gateway), and Healthcare IoT Sensor Node (Sensor Node).

1.Medical Service Gateway (Gateway): Since the user and the sensor node do not communicate with each other directly, the gateway is responsible for authentication and passing the communication between the user and the sensor node.2.Healthcare IoT Sensor Node (Sensor Node): The Healthcare IoT Sensor Node is attached to the patient’s body and collects the patient’s medical data. The Healthcare IoT Sensor Node is connected to the healthcare network and transmits the patient’s medical data to the user through a Gateway.3.Doctor (User): The user is a doctor who can access the patient’s medical information that has been collected from the sensor node to inform the patient’s treatment.

### 3.3. Attack Model

For security analysis in our scheme, we consider the following attack model [20,21,22]:1.The attacker can extract the data in the device that stores some security parameters.2.The attacker can access the public communication channel, at which point the attacker can interrupt, return, amend and eliminate or transmit the message.3.The attacker can calculate the identity and password in polynomial time.

## 4. Review of the Scheme Presented by Masud et al. [6]

In this section, we briefly review Masud et al.’s scheme [6] that only uses password protection to only let legitimate users access the IoT sensor node to obtain the patient’s health information. The notations and Masud et al. [6]’s scheme are described in Table 2 and Table 3, respectively.

Masud et al. [6]’s scheme consists of three phases: the User Registration Phase, the Sensor Node Registration Phase, and the Mutual Authentication and Key Agreement Phase.

### 4.1. User Registration Phase

1.The user enters and transmits $D_{ID}$ and $PW_{D}$ through the secured channel to the gateway.2.The gateway generates $R_{SG}^{1}$ and computes $D_{TID}=R_{SG}^{1}⊕D_{ID}$ and $\alpha =\left(D_{ID}⊕R_{SG}^{1}\right)⊕PW_{D}$. The gateway stores $D_{ID}$, $PW_{D}$, $R_{SG}^{1}$ and $D_{TID}$. Finally, the gateway transmits α through the secured channel to the user.3.The user derives $R_{SG}^{1*}=\left(\alpha ⊕PW_{D}\right)⊕D_{ID}$ and computes $D_{TID}=R_{SG}^{1*}⊕D_{ID}$ and $\beta =h\left(PW_{D}\|R_{SG}^{1*}\right)⊕D_{TID}$. The user then stores $R_{SG}^{1*},D_{TID}$, and β.

### 4.2. Sensor Node Registration Phase

1.The sensor node generates $R_{SN}^{1}$ and then transmits $R_{SN}^{1}$ and $S_{ID}$ through the secured channel to the gateway.2.The gateway generates $R_{SG}^{2}$ and computes $\delta =(S_{ID}$⊕$R_{SG}^{2})⊕R_{SN}^{1}$ and $S_{TID}=R_{SG}^{2}$⊕$S_{ID}$. The gateway then stores $S_{ID},R_{SN}^{1},R_{SG}^{2}$, and $S_{TID}$. Finally, the gateway transmits δ through the secured channel to the sensor.3.The sensor node computes $R_{SG}^{2*}=\left(\delta ⊕R_{SN}^{1}\right)⊕S_{ID}$ and $S_{TID}=R_{SN}^{2*}⊕S_{ID}$. The sensor node then stores $R_{SN}^{1},R_{SG}^{2*},S_{TID}$.

### 4.3. Mutual Authentication and Key Agreement Phase

1.The user enters $PW_{D}$ and then computes $Q=h\left(PW_{D}\|R_{SG}^{1*}\right)⊕D_{TID}$, where $R_{SG}^{1*}$ and $D_{TID}$. If *Q* and β are not equal, the procedure is stopped; if they are equal, the user generates $N_{D}^{1}$ and computes $N_{D}^{1*}=N_{D}^{1}⊕PW_{D}$ and $\lambda =h(R_{SG}^{1*}\|PW_{D}$). Finally, the user transmits $N_{D}^{1*},D_{TID},\lambda$, and $S_{TID}$ to the gateway.2.The gateway retrieves $N_{D}^{1}=N_{D}^{1*}⊕PW_{D}$, after which the gateway checks the freshness of $N_{D}^{1}$. If $N_{D}^{1}$ is not fresh, the procedure is stopped; else the gateway compares $D_{TID}$ and $S_{TID}$ with the received values. If the values are not identical, the procedure is stopped; else the gateway computes $\lambda^{*}=h\left(R_{SG}^{1}\|PW_{D}\right)$. If $\lambda^{*}$ and λ are equal, the user authentication is successful. If they are not equal, the procedure is stopped. To share SK with the sensor node, the gateway generates $N_{G}^{1}$, SK, and $R_{SG}^{3}$. The gateway computes $G_{W}^{1}=N_{G}^{1}⊕S_{TID}$, $G_{W}^{2}=h\left(R_{SN}^{1}\|R_{SG}^{2}\right)$, $SK_{S}=\left(SK⊕R_{SN}^{1}\right)⊕N_{G}^{1}$ and $G_{W}^{3}=R_{SG}^{3}⊕R_{SN}^{1}$. Finally, the gateway stores $G_{W}^{3}$ and transmits $G_{W}^{1}$, $G_{W}^{2}$, $D_{TID}$, $SK_{S}$, and $G_{W}^{3}$ through the public channel to the sensor node.3.The sensor node retrieves $N_{G}^{1}=G_{W}^{1}⊕S_{TID}$ and then checks the freshness of $N_{G}^{1}$. If $N_{G}^{1}$ is not fresh, the procedure is stopped; else the sensor node computes $S_{N}^{1}=h\left(R_{SN}^{1}\|R_{SG}^{2*}\right)$. If $S_{N}^{1}$ and $G_{W}^{2}$ are equal, the gateway authentication is successful. Then, the sensor node retrieves $SK=\left(SK_{S}⊕R_{SN}^{1}\right)⊕N_{G}^{1}$ and the sensor node generates $N_{S}^{1}$ and $R_{SN}^{2}$. The sensor node then computes $S_{N}^{2}=N_{S}^{1}⊕S_{TID}$, $S_{N}^{3}=h(R_{SG}^{2*}\|R_{SN}^{1}\left\|SK\right)$, and $S_{N}^{4}=R_{SG}^{2*}⊕R_{SN}^{2}$. Moreover, the sensor node retrieves $R_{SG}^{3}=G_{W}^{3}⊕R_{SN}^{1}$ and derives $S_{TID}^{new}=R_{SG}^{3}⊕S_{ID}$. The sensor node stores $R_{SN}^{2},R_{SG}^{3},S_{TID}^{new},$ and $S_{N}^{3}$. Finally, the sensor node transmits $S_{N}^{2},S_{N}^{3},$ and $S_{N}^{4}$ to the gateway.4.The gateway retrieves $N_{S}^{1}=S_{N}^{2}⊕S_{TID}$ and $R_{SN}^{2}=S_{N}^{4}⊕R_{SG}^{2}$. The gateway then checks the freshness of $N_{S}^{1}$. If $N_{S}^{1}$ is not fresh, the procedure is stopped; else the gateway computes $G_{W}^{4}=h(R_{SG}^{2}\|R_{SN}^{1}\left\|SK\right)$ and $S_{TID}^{new}=S_{SG}^{3}⊕S_{ID}$. If $G_{W}^{4}$ and $S_{N}^{3}$ are equal, the mutual authentication between the sensor node and the gateway is successful. Subsequently, the gateway stores $R_{SN}^{2}$, $R_{SG}^{3}$, and $S_{TID}^{new}$. To share SK with the user, the gateway generates $N_{G}^{2}$ and $R_{SG}^{4}$. The gateway then computes $\mu =D_{ID}⊕N_{G}^{2}$, $SK_{U}=\left(SK⊕PW_{D}\right)⊕N_{G}^{2}$, $\eta =h(D_{ID}\|PW_{D}\|SK\|N_{G}^{2})$, $G_{W}^{5}=R_{SG}^{4}⊕PW_{D}$ and $D_{TID}^{new}=R_{SG}^{4}⊕D_{ID}$. Finally the gateway stores $R_{SG}^{4}$ and $D_{TID}^{new}$ and ultimately transmits μ, $SK_{u}$, η, and $G_{W}^{5}$ to the user.5.The user retrieves $N_{G}^{2}=\mu ⊕D_{ID}$. The user then checks the freshness of $N_{G}^{2}$. If $N_{G}^{2}$ is not fresh, the procedure is stopped; else the user computes $\phi =h(D_{ID}\|PW_{D}\|SK\|N_{G}^{2})$. If ϕ and η are equal, the mutual authentication between the gateway and the user is successful. Then the user retrieves $SK=\left(SK_{U}⊕N_{G}^{2}\right)⊕PW_{D}$ and $R_{SG}^{4}=G_{W}^{5}⊕PW_{D}$. The user computes $D_{TID}^{new}=R_{SG}^{4}⊕D_{ID}$ and stores $R_{SG}^{4}$ and $D_{TID}^{new}$.

## 5. Weaknesses of Masud et al. [6]’s Scheme

These weaknesses are listed under the assumption that the attacker has recorded the message $\left(N_{D}^{1*},D_{TID},\lambda ,S_{TID}\right)$ from a successful mutual authentication and key agreement of the user *A*.

### 5.1. Offline Password Guessing Attack

In an offline password guessing attack, the attacker is never actually attempting to login to the gateway server. Suppose the attacker steals the device of user *A* and obtains $R_{SG}^{1*}$ from the device. Then, the attacker repeatedly guesses a password $PW_{D}^{*}$ and computes $\lambda^{*}=h\left(R_{SG}^{1*}\|PW_{D}^{*}\right)$ offline. If $\lambda^{*}$ is equal to λ, the attacker can obtain the correct password $PW_{D}$. Until the attacker determines a valid user’s password $PW_{D}$, the gateway does not notice this attack at all because the attacker does not try to login.

Then, in the Mutual Authentication and Key Agreement Phase, the attacker retrieves $D_{ID}=R_{SG}^{1*}⊕D_{TID}$, $N_{G}^{2}=\mu ⊕D_{ID}$ and $SK=\left(SK_{U}⊕N_{G}^{2}\right)⊕PW_{D}$. Eventually, the attacker can obtain an SK that can be used to access the resource of the gateway and the sensor node.

### 5.2. Privileged Insider Attack

If a privileged insider of the gateway has obtained the user *A*’s password $PW_{D}$, $D_{ID}$ and $R_{SG}^{1}$ from the gateway’s database, he is trying to impersonate user *A*. In the Mutual Authentication and Key Agreement Phase, the privileged insider retrieves $N_{G}^{2}=\mu ⊕D_{ID}$ and $SK=\left(SK_{U}⊕N_{G}^{2}\right)⊕PW_{D}$. Eventually, the privileged insider can obtain an SK that can access the resource of the gateway and the sensor node.

### 5.3. Replay Attack

Masud et al. [6] claim their scheme is safe from replay attacks because the gateway checks the freshness of nonce $N_{D}^{1*}=N_{D}⊕PW_{D}$ and that the nonce cannot be modified since it is secretly enclosed in the password $PW_{D}$. However, suppose the attacker generates the nonce $N_{A}$ and then transmits $\left(N_{A},D_{TID},\lambda ,S_{TID}\right)$ instead of $\left(N_{D},D_{TID},\lambda ,S_{TID}\right)$. Upon receiving $\left(N_{A},D_{TID},\lambda ,S_{TID}\right)$, the gateway retrieves $N_{D}^{1*}=N_{A}⊕PW_{D}$ and then the gateway checks the freshness of $N_{D}^{1*}$. However, since the retrieved $N_{D}^{1*}$ is a random number if only freshness is guaranteed, the gateway cannot confirm whether $N_{D}^{1*}$ is valid. Briefly, if the attacker can make a nonce that can guarantee freshness, Masud et al. [6]’s scheme cannot resist replay attacks. Further, if freshness is proven since the new identities of user and device $D_{TID}^{new}$, $S_{TID}^{new}$ are changed, valid users who do not know the changed identity $D_{TID}^{new}$ will no longer be able to authenticate themselves after the replay attack.

## 6. Proposed Scheme

In this section, we propose a three-factor mutual authentication scheme for the Internet of Medical Things (IoMT) that is intended to overcome the weaknesses of the scheme reported by Masud and colleagues. The proposed scheme only consists of three phases: user registration, sensor node registration, and authentication and key distribution. The proposed scheme is described in Table 4.

### 6.1. User Registration Phase

1.The user enters $ID_{i}$ and $PW_{D}$ and generates a random number $r_{U}^{1}$. The user imprints $B_{i}$ on a device for biometric collection and computes $Gen\left(B_{i}\right)=\left(R_{i},R_{bi}\right)$ and $HPW_{i}=h(PW_{i}\|R_{i}\left\|r_{U}^{1}\right)$. For registration, the user transmits $ID_{i}$ through a secured channel to the gateway.2.The gateway generates random numbers $r_{GW}^{1}$, $r_{GW}^{2}$ and $r_{GW}^{3}$ and computes $TID_{i}=h(ID_{i}\|r_{GW}^{1}\left\|K_{GW}\right)$, $S_{i}^{1}=h(ID_{i}\|r_{GW}^{2}\left\|K_{GW}\right)$ and $S_{i}^{2}=h(ID_{i}\|r_{GW}^{3}\left\|K_{GW}\right)$. The gateway stores $ID_{i}$, $TID_{i}$, $S_{i}^{1}$ and $S_{i}^{2}$. $S_{i}^{2}$ is temporarily stored by the gateway until the sensor node registration phase, is transferred from the gateway to the sensor node during the sensor node registration phase, and is then deleted from the gateway. Finally, the gateway transmits $TID_{i}$, $S_{i}^{1}$, and $S_{i}^{2}$ through a secured channel to the user.3.The user computes $U_{i}M_{1}=TID_{i}⊕HPW_{i}$, $U_{i}M_{2}=S_{i}^{1}⊕HPW_{i}$ and $U_{i}M_{3}=S_{i}^{2}⊕HPW_{i}$, $U_{i}M_{4}=h(PW_{i}\|R_{i}\left\|ID_{i}\right)⊕r_{U}^{1}$ and $U_{i}M_{5}=h(r_{U}^{1}\|TID_{i}\|S_{i}^{1}\left\|S_{i}^{2}\right)$ and stores $U_{i}M_{1}$, $U_{i}M_{2}$, $U_{i}M_{3}$, $U_{i}M_{4}$, and $U_{i}M_{5}$.

### 6.2. Sensor Node Registration Phase

1.For registration, the sensor node transmits $SID_{j}$ through a secured channel to the gateway.2.The gateway generates a random number $r_{GW}^{4}$ and computes $TSID_{j}=h(SID_{j}\|r_{GW}^{4}\left\|K_{GW}\right)$ and stores $SID_{j}$, $TSID_{j}$. Finally, the gateway transmits $TSID_{j}$, $TID_{i}$, and $S_{i}^{2}$ through a secured channel to the sensor node and deletes $S_{i}^{2}$.3.The sensor node stores $SID_{j}$, $TSID_{j}$, $TID_{i}$, and $S_{i}^{2}$.

### 6.3. Mutual Authentication and Key Distribution Phase

1.The user enters $ID_{i}$ and $PW_{i}$ and imprints $B_{i}$ on a device for biometric collection and computes $R_{i}=Rep\left(B_{i},R_{bi}\right)$, $r_{U}^{1}=h(PW_{i}\|R_{i}\left\|ID_{i}\right)⊕U_{i}M_{4}$, $HPW_{i}=h(PW_{i}\|R_{i}\left\|r_{U}^{1}\right)$, $TID_{i}=U_{i}M_{1}⊕HPW_{i}$, $S_{i}^{1}=U_{i}M_{2}⊕HPW_{i}$, $S_{i}^{2}=U_{i}M_{3}⊕HPW_{i}$ and $U_{i}M_{5}^{*}=h(r_{U}^{1}\|TID_{i}\|S_{i}^{1}\left\|S_{i}^{2}\right)$. To check the user’s password, the user checks if $U_{i}M_{5}$ = ? $U_{i}M_{5}^{*}$. If the equation is equal, the password check is passed; if not, the procedure is stopped. To generate an authentication message, the user generates $r_{U}^{2}$, $r_{U}^{3}$ and $ts_{U}$ and computes $U_{i}M_{6}=r_{U}^{2}⊕S_{i}^{1}$, $U_{i}M_{7}=r_{U}^{3}⊕S_{i}^{2}$ and $U_{i}M_{8}=h(r_{U}^{2}\|TID_{i}\left\|ts_{U}\right)$. Finally the user transmits $U_{i}M_{6}$, $U_{i}M_{7}$, $U_{i}M_{8}$, $TID_{i}$ and $ts_{U}$ through a public channel to the gateway.2.To authenticate the user, the gateway retrieves $r_{U}^{2}=U_{i}M_{6}⊕S_{i}^{1}$ and computes $U_{i}M_{8}^{*}=h(r_{U}^{2}\|TID_{i}\left\|ts_{U}\right)$. The gateway checks $r_{U}^{2}$’s freshness and if $U_{i}M_{8}$= ? $U_{i}M_{8}^{*}$. The gateway also checks for whether or not $ts_{U}$ is a valid range. If $U_{i}M_{8}^{*}$ and $ts_{U}$ are valid, the user verification is passed; if not, the procedure is stopped. To generate the authentication message, the gateway generates $r_{GW}^{5}$ and computes $TID_{i}^{new}=h(ID_{i}\|r_{GW}^{5}\left\|K_{GW}\right)$, $GM_{1}=TSID_{j}⊕r_{GW}^{5}$, $GM_{2}=TID_{i}⊕r_{GW}^{5}$, $GM_{3}=TID_{i}^{new}⊕r_{GW}^{5}$ and $GM_{4}=h(r_{GW}^{5}\|TSID_{j}\|TID_{i}\left\|TID_{i}^{new}\right)$. Finally the gateway transmits $GM_{1}$, $GM_{2}$, $GM_{3}$, $GM_{4}$, and $U_{i}M_{7}$ through a public channel to the sensor node.3.To authenticate the gateway, the sensor node retrieves $r_{GW}^{5}=GM_{1}⊕TSID_{j}$, $TID_{i}=GM_{2}⊕r_{GW}^{5}$, $TID_{i}^{new}=GM_{3}⊕r_{GW}^{5}$ and $r_{U}^{3}=U_{i}M_{7}⊕S_{i}^{2}$ then computes $GM_{4}^{*}=h(r_{GW}^{5}\|TSID_{j}\|TID_{i}\left\|TID_{i}^{new}\right)$. The sensor node checks $GM_{4}$ = ? $GM_{4}^{*}$. If the equation is equal, the gateway verification is passed; if not, the procedure is stopped. To generate the session key SK, the sensor node generates SK and computes $SN_{j}M_{1}=SK⊕S_{i}^{2}$, $SN_{j}M_{2}=h(SK\|TID_{i}^{new}\left\|r_{U}^{3}\right)$ and $SN_{j}M_{3}=h(SN_{j}M_{1}\|SN_{j}M_{2}\left\|TSID_{j}\right)$. The sensor node replaces $TID_{i}$ by $TID_{i}^{new}$. Finally, the sensor node transmits $SN_{j}M_{1}$, $SN_{j}M_{2}$ and $SN_{j}M_{3}$ through a public channel to the gateway.4.To authenticate the sensor node, the gateway computes $SN_{j}M_{3}^{*}=h(SN_{j}M_{1}\|SN_{j}M_{2}\left\|TSID_{j}\right)$ and checks if $SN_{j}M_{3}$ = ? $SN_{j}M_{3}^{*}$. If the equation is equal, the sensor node verification is passed. To generates an authentication message, the gateway computes $GM_{5}=h\left(TID_{i}\|TID_{i}^{new}\right)$ and $GM_{6}=TID_{i}^{new}⊕S_{i}^{1}$ and replaces $TID_{i}$ by $TID_{i}^{new}$. Finally, the gateway transmits $GM_{5}$, $GM_{6}$, $SN_{j}M_{1}$ and $SN_{j}M_{2}$ through a public channel to the user.5.To authenticate the gateway, the user computes $TID_{i}^{new}=GM_{6}⊕S_{i}^{1}$, $GM_{5}^{*}=h\left(TID_{i}\|TID_{i}^{new}\right)$ and $U_{i}M_{1}^{new}=TID_{i}^{new}⊕HPW_{i}$. The user checks if $GM_{5}$ = ? $GM_{5}^{*}$. If the equation is equal, the gateway verification is passed; if not, the procedure is stopped. To obtain the session key SK, the user retrieves $SK=SN_{j}M_{1}⊕S_{i}^{2}$ and computes $SN_{j}M_{2}^{*}=h(SK\|TID_{i}^{new}\left\|r_{U}^{3}\right)$. The user checks $SN_{j}M_{2}$ = ? $SN_{j}M_{2}^{*}$. If the equation is equal, the user obtains the valid session key SK; if not, the procedure is stopped.

## 7. Security Analysis of the Proposed Scheme

In this section, we demonstrate formal and informal security analysis. We use the security verification tool ProVerif to demonstrate that the proposed scheme can satisfy security and authentication features. As an informal security analysis, we show how our proposed scheme meets the security requirements for an IoMT sensor protocol.

### 7.1. Formal Security Analysis

In this section, the ProVerif tool [23] is used to evaluate the security of the proposed protocol. ProVerif tool is an automatic cryptographic protocol verifier that was developed by Bruno Blanchet [13]. Several studies have used this tool to demonstrate the safety of their protocols [24,25].

We use two types, and four channels in total. Private channel1 and Private channel2 transmit sensitive data between the user and the gateway and between the gateway and the sensor node, respectively. Public channel1 and Public channel2 transmit general data between the user and the gateway and between the gateway and the sensor node, respectively. Table 5 presents the definitions of the channels, variables, and other related parameters. The processes performed by the user, the gateway, and the sensor node are presented in Table 6, Table 7 and Table 8, respectively. Lastly, the queries and main process are detailed in Table 9.

The results of our proposed scheme are presented in Table 10. It can be seen that the proposed protocol kept the session key SK safe from the attacker.

When we run the following query in Table 9, we can obtain the following results:1.Query inj–event(endEVENTA) ==> inj-event(startEVENTA) is true.2.Query inj–event(endEVENTB) ==> inj-event(startEVENTB) is false.3.Query not attacker(M) is true.4.Query not attacker(M) is false.

“Query inj-event (endEVENTA) == > inj-event (startEVENTA) is true.” means that the process from endEVENTA to startEVENTA has been authenticated. By contrast, “Query inj-event (endEVENTB) == > inj-event (startEVENTB) is false.” means that the authentication from endEVENTB to startEVENTB is not successful. “Query not attacker (M) is true.” means that an attacker cannot acquire a free name M. Finally, “Query not attacker (M) is false.” means that an attacker can trace the M.

The query results from Table 9 are listed in Table 10.

### 7.2. Informal Security Analysis

We performed a formal analysis. However, a formal analysis by itself is not sufficient to prove safety [13,26,27]. Therefore, we further analyzed our scheme using an informal analysis. We present a theoretical analysis of the proposed scheme. The results of the informal security analysis are then briefly described.

1.Offline Password Guessing Attack: Since our scheme uses biometric information $B_{i}$ with the unique biological characteristics of individuals that are not stored for user authentication, it is impossible to guess a user’s password without a real user. Therefore our scheme can protect against the offline password guessing attack.2.Privileged Insider Attack: Even if the privileged insider steals $ID_{i}$, $TID_{i}$, $SID_{j}$, $TSID_{j}$, and $S_{i}^{1}$ from the gateway’s database, the privileged insider can not obtain the session key SK without secret information $S_{i}^{2}$ that is shared between the user and the sensor node. Therefore our scheme can protect against privileged insider attacks.3.User Impersonation Attack: Even if the attacker steals and replaces the user’s $TID_{i}$, the attacker can not generate valid $U_{i}M_{6}$ and $U_{i}M_{7}$ without secret information $S_{i}^{1}$ and $S_{i}^{2}$. When the gateway and the sensor node verify $U_{i}M_{6}$ and $U_{i}M_{7}$, respectively, they can find the invalid user. Therefore our scheme can protect against user impersonation attacks.4.Server Impersonation Attack: Even if the attacker impersonates the gateway, the attacker does not generate valid $GM_{4}$ and $GM_{5}$ without $TSID_{j}$ and $S_{i}^{1}$. When the sensor node and the user verify $GM_{4}$ and $GM_{5}$, respectively, they can find the invalid gateway. Therefore our scheme can protect against server impersonation attacks.5.Replay Attack: Even if the attacker steals $U_{i}M_{6}$, $U_{i}M_{7}$, $U_{i}M_{8}$, and $TID_{i}$ from a successful mutual authentication and key distribution phase and then resends it to the gateway, the gateway can find whether or not the message is reused because the gateway checks $r_{U}^{2}$’s freshness. Moreover, the attacker can not generate and modify $r_{U}^{2}$ and $U_{i}M_{6}$ without $S_{i}^{1}$. Therefore our scheme can protect against replay attacks.6.Man-in-the-Middle Attack: In a man-in-the-middle attack, an attacker puts themselves in the middle of two parties so that they can intercept and modify some communicated data to masquerade as the entities. In the mutual authentication and key distribution phase, the attacker intercepts communicated data between the user and the gateway and attempts to modify the message to retrieve the session key. However, in our scheme, the attacker can not modify communicated messages without the secret information $S_{i}^{1}$ and $S_{i}^{2}$. Therefore our scheme can protect against man-in-the-middle attacks.7.Session Key Disclosure Attack: Even if the attacker obtains $SN_{j}M_{1}$ which includes the session key SK, the attacker can not obtain the session key without the secret information $S_{i}^{2}$. Therefore, our scheme can protect against session key disclosure attacks.8.Forward Secrecy and Backward Secrecy: Even if someone gains the session key SK, they can not know the old session key or the new session key because each session key is generated randomly with no relation to the other session keys. Therefore our scheme can preserve forward secrecy and backward secrecy.9.Mutual Authentication: The gateway and the user can authenticate each other by verifying $U_{i}M_{8}$ and $GM_{5}$ respectively, using the secret information $S_{i}^{1}$. Therefore our scheme provides mutual authentication.10.User Anonymity: Our scheme identifies users using $TID_{i}$ and then replaces it every time with $TID_{i}^{new}$ regardless of the old $TID_{i}$. Therefore our scheme preserves user anonymity.

The results of the security analysis with comparisons to related papers are presented in Table 11.

## 8. Performance Analysis of the Proposed Scheme

Many authentication studies have analyzed their performance in the following manner [9,10,11,12,13,14,15,20,21,22,24,25,26,27,28,29,30,31,32]. We compared each computation amount in terms of the research methods used. We analyze the performance of our scheme as follows.

Our study analyzes the computational cost using the time measurement presented in Table 12 [28,29]. $T_{M}$ stands for the computational cost of multiplication in the field. $T_{bh}$ stands for the computational cost of the biohash function operation, and $T_{h}$ stands for the computational cost of the one-way hash function operation. It is assumed that the XOR operation does not affect the cost of operation. Table 13 and Figure 2 compare the computational cost of our scheme with those of other schemes according to Table 12 [6,12,15].

We calculate the computational efficiency of our scheme as follows:(3)$$\left(t_{1}-t_{2}\right)/t_{2}$$

In Formula (Equation 3), $t_{1}$ represents the average cost of computation of the different schemes. Moreover, $t_{2}$ represents the cost of operation of our scheme.

According to the above formula, the operation of our scheme is 266.48% more efficient in terms of computational cost than the other schemes, and Table 11 shows that our scheme is more secure than the other methods.

## 9. Discussion of Performance

We proposed a secure and lightweight user authentication scheme for IoMT by improving Masud et al. [6]’s scheme. We compared the performance of three schemes [6,12,15] in Section 8. Our scheme outperforms [12,15] by 399.73% and 499.66% respectively. The performance of [6] is lightweight, but it does not meet basic security requirements such as offline password guessing attacks, privileged insider attacks, and replay attacks. Therefore, our scheme is a suitable lightweight user authentication scheme for IoMT because our scheme not only is improved by addressing the security threats of [6] but also outperforms 266.48% more efficiently than the other schemes.

## 10. Conclusions

The purpose of our paper was to propose a secure and lightweight user authentication scheme for IoMT by addressing the security threats to which Masud et al. [6]’s scheme is vulnerable. In particular, our scheme can protect against well-known attacks in IoMT i.e., offline password guessing attacks, privileged insider attacks, user impersonation attacks, replay attacks, and session key disclosure attacks, and it ensures user anonymity. We also proved that our scheme is a suitable user authentication scheme for IoMT through formal security analysis by ProVerif. Moreover, we proposed a lightweight security protocol that mainly uses a hash function and XOR operation considering low-spec healthcare sensors. As a result, we showed 266.48% better performance than the average computational cost of the considered schemes [6,12,15]. Our scheme outperforms [12,15], but it does not outperform [6]. Our scheme shows higher safety than the compared schemes [6,12,15]. Our security and performance analysis shows that our scheme is a suitable lightweight user authentication scheme for IoMT. Further studies will be able to improve convenience by combining behavioral biometrics authentication. Behavioral biometric authentication is expected to achieve further improved convenience over biometrics authentication because it uses keystroke dynamics, gait analysis, mouse use characteristics, signature analysis, and cognitive biometrics.

## Figures and Tables

**Figure 1 sensors-23-01122-f001:**
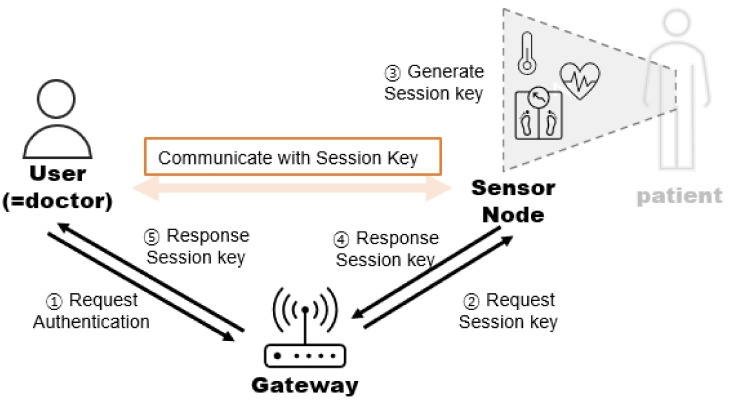
The system model of our scheme in IoMT.

**Figure 2 sensors-23-01122-f002:**
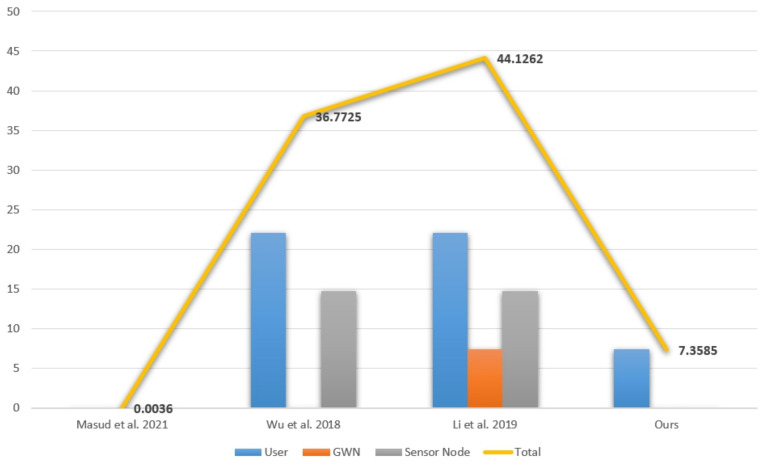
Graph comparisons of computational cost [6,12,15].

**Table 1 sensors-23-01122-t001:** Summary of related works.

Author	Proposed Scheme	Weakness
Wu et al. [9]	For TMIS using a smart card	Impersonation, insider, offline password guessing attacks
Debiao et al. [10]	Using a smart card and a password	Offline password guessing attacks
Wu et al. [12]	Secure three-factor scheme for wireless sensor networks	User impersonation attacks and no user anonymity
Masud et al. [6]	For IoT-based healthcare	Offline-password, replay, privileged insider attacks

**Table 2 sensors-23-01122-t002:** Notations.

Notion	Description
$U_{i}M_{*}$, $GM_{*}$, $SN_{i}M_{*}$	*-th message of *i*-th user, gateway, and *j*-th sensor node, respectively
$ID_{i}$, $SID_{j}$	Identity of *i*-th user, *j*-th sensor node, respectively
$PW_{i}$	Password of *i*-th user
$B_{i}$	Biometric Information of *i*-th user
$r_{U}^{*}$, $r_{GW}^{*}$, $r_{SN}^{*}$	*-th random number generated by user, gateway, and sensor node, respectively
$ts_{U}$	Timestamp of user
$S_{i}^{1}$	Secret information between user and gateway
$S_{i}^{2}$	Secret information between user and sensor node
$D_{ID}$, $S_{ID}$	Identity of user, sensor node
$PW_{D}$	Device password set by doctor
$R_{SG}$, $R_{SN}$	Random secret generated by gateway, sensor node, respectively
$N_{D}$, $N_{G}$, $N_{S}$	Nonce generated by user’s device, gateway, sensor node, respectively
$K_{GW}$	Secret key of gateway
h(.)	Hash function
‖	Concatenation operator
⊕	Bit wise XOR
SK	Session key

**Table 3 sensors-23-01122-t003:** Masud et al.’s [6] scheme.

User	Gateway	Sensor Node
{User Registration Phase}		
Enter: $D_{ID}$, $PW_{D}$		
Transmit: $D_{ID}$, $PW_{D}$ ⟶	Generate: $R_{SG}^{1}$	
	Compute: $D_{TID}=R_{SG}^{1}⊕D_{ID}$	
	$\alpha =\left(D_{ID}⊕R_{SG}^{1}\right)⊕PW_{D}$	
	Store: $D_{ID}$, $PW_{D}$, $R_{SG}^{1}$, $D_{TID}$	
Derives: $R_{SG}^{1*}=\left(\alpha ⊕PW_{D}\right)⊕D_{ID}$	⟵ Transmit: α	
Compute: $D_{TID}=R_{SG}^{1*}⊕D_{ID}$,		
$\beta =h\left(PW_{D}\|R_{SG}^{1*}\right)⊕D_{TID}$		
Store: $R_{SG}^{1*},D_{TID}$, β		
{Sensor Node Registration Phase}		
		Generate: $R_{SN}^{1}$
	Generate: $R_{SG}^{2}$	⟵ Transmit: $R_{SN}^{1}$, $S_{ID}$
	Compute: $\delta =\left(S_{ID}⊕R_{SG}^{2}\right)⊕R_{SN}^{1}$,	
	$S_{TID}=R_{SG}^{2}⊕S_{ID}$	
	Store: $S_{ID},R_{SN}^{1},R_{SG}^{2}$, $S_{TID}$	
	Transmit: δ ⟶	Compute: $R_{SG}^{2*}=\left(\delta ⊕R_{SN}^{1}\right)⊕S_{ID}$,
		$S_{TID}=R_{SN}^{2*}⊕S_{ID}$
		Store: $R_{SN}^{1},R_{SG}^{2*},S_{TID}$
{Mutual Authentication and Key agreement}		
**(1)**		
Enter: $PW_{D}$	**(2)**	
Compute: $Q=h\left(PW_{D}\|R_{SG}^{1*}\right)⊕D_{TID}$	Retrieves: $N_{D}^{1}=N_{D}^{1*}⊕PW_{D}$	
Verify: *Q* = ? β	Verify: $N_{D}^{1}$’s freshness	
Generate: $N_{D}^{1}$	Check: $D_{TID}$, $S_{TID}$ are stored	
Compute: $N_{D}^{1*}=N_{D}^{1}⊕PW_{D}$,	Compute: $\lambda^{*}=h\left(R_{SG}^{1}\|PW_{D}\right)$	
$\lambda =h(R_{SG}^{1*}\|PW_{D})$	Verify: $\lambda^{*}$ = ? λ	
Transmit: $N_{D}^{1*},D_{TID},\lambda$, $S_{TID}$ ⟶	Generate: $N_{G}^{1}$, SK, $R_{SG}^{3}$	
	Compute: $G_{W}^{1}=N_{G}^{1}⊕S_{TID}$,	**(3)**
	$G_{W}^{2}=h\left(R_{SN}^{1}\|R_{SG}^{2}\right)$,	Retrieve: $N_{G}^{1}=G_{W}^{1}⊕S_{TID}$
	$SK_{S}=\left(SK⊕R_{SN}^{1}\right)⊕N_{G}^{1}$,	Verify: $N_{G}^{1}$’s freshness
	$G_{W}^{3}=R_{SG}^{3}⊕R_{SN}^{1}$	Compute: $S_{N}^{1}=h\left(R_{SN}^{1}\|R_{SG}^{2*}\right)$
	Store: $G_{W}^{3}$	Verify: $S_{N}^{1}$ = ? $G_{W}^{2}$
	Transmit: $G_{W}^{1}$, $G_{W}^{2}$, $D_{TID}$,	Retrieve: $SK=\left(SK_{S}⊕R_{SN}^{1}\right)⊕N_{G}^{1}$
	$SK_{S}$, $G_{W}^{3}$⟶	Generate: $N_{S}^{1}$, $R_{SN}^{2}$
		Retrieve: $R_{SG}^{3}=G_{W}^{3}⊕R_{SN}^{1}$
	**(4)**	Compute: $S_{N}^{2}=N_{S}^{1}⊕S_{TID}$,
	Retrieve: $N_{S}^{1}=S_{N}^{2}⊕S_{TID}$,	$S_{N}^{3}=h(R_{SG}^{2*}\|R_{SN}^{1}\left\|SK\right)$,
	$R_{SN}^{2}=S_{N}^{4}⊕R_{SG}^{2}$	$S_{N}^{4}=R_{SG}^{2*}⊕R_{SN}^{2}$,
	Verify: $N_{S}^{1}$’s freshness	$S_{TID}^{new}=R_{SG}^{3}⊕S_{ID}$
	Compute: $G_{W}^{4}=h(R_{SG}^{2}\|R_{SN}^{1}\left\|SK\right)$,	Store: $R_{SN}^{2},R_{SG}^{3},S_{TID}^{new},S_{N}^{3}$
	$S_{TID}^{new}=S_{SG}^{3}⊕S_{ID}$	⟵ Transmit: $S_{N}^{2},S_{N}^{3},S_{N}^{4}$
	Verify: $G_{W}^{4}$ = ? $S_{N}^{3}$	
**(5)**	Store: $R_{SN}^{2}$, $R_{SG}^{3}$, $S_{TID}^{new}$	
Retrieve: $N_{G}^{2}=\mu ⊕D_{ID}$	Generate: $N_{G}^{2}$, $R_{SG}^{4}$	
Verify: $N_{G}^{2}$’s freshness	Compute: $\mu =D_{ID}⊕N_{G}^{2}$,	
Compute: $\phi =h(D_{ID}\|PW_{D}\|SK\|N_{G}^{2})$	$SK_{U}=\left(SK⊕PW_{D}\right)⊕N_{G}^{2}$,	
Verify: ϕ = ? η	$\eta =h(D_{ID}\|PW_{D}\|SK\|N_{G}^{2})$,	
Retrieve: $SK=\left(SK_{U}⊕N_{G}^{2}\right)⊕PW_{D}$,	$G_{W}^{5}=R_{SG}^{4}⊕PW_{D}$,	
$R_{SG}^{4}=G_{W}^{5}⊕PW_{D}$	$D_{TID}^{new}=R_{SG}^{4}⊕D_{ID}$	
Compute: $D_{TID}^{new}=R_{SG}^{4}⊕D_{ID}$	Store:$R_{SG}^{4}$, $D_{TID}^{new}$	
Store: $R_{SG}^{4}$, $D_{TID}^{new}$	⟵ Transmit:μ, $SK_{u}$, η, and $G_{W}^{5}$	

**Table 4 sensors-23-01122-t004:** The proposed scheme.

User($U_{i}$)	Gateway	Sensor Node(${\mathrm{SN}}_{j}$)
{User Registration}		
**(1)** Enter: $ID_{i}$, $PW_{i}$, $B_{i}$		
Generate: $r_{U}^{1}$	**(2)** Generate: $r_{GW}^{1}$, $r_{GW}^{2}$, $r_{GW}^{3}$	
Compute: $Gen\left(B_{i}\right)=\left(R_{i},R_{bi}\right)$,	Compute:	
$HPW_{i}=h(PW_{i}\|R_{i}\left\|r_{U}^{1}\right)$	$TID_{i}=h(ID_{i}\|r_{GW}^{1}\left\|K_{GW}\right)$,	
Transmit: $ID_{i}$ ⟶	$S_{i}^{1}=h(ID_{i}\|r_{GW}^{2}\left\|K_{GW}\right)$,	
	$S_{i}^{2}=h(ID_{i}\|r_{GW}^{3}\left\|K_{GW}\right)$	
**(3)** Compute:	Store: $ID_{i}$, $TID_{i}$, $S_{i}^{1}$, $S_{i}^{2}$	
$U_{i}M_{1}=TID_{i}⊕HPW_{i}$,	⟵ Transmit: $TID_{i}$, $S_{i}^{1}$, $S_{i}^{2}$	
$U_{i}M_{2}=S_{i}^{1}⊕HPW_{i}$		
$U_{i}M_{3}=S_{i}^{2}⊕HPW_{i}$		
$U_{i}M_{4}=h(PW_{i}\|R_{i}\left\|ID_{i}\right)⊕r_{U}^{1}$		
$U_{i}M_{5}=h(r_{U}^{1}\|TID_{i}\|S_{i}^{1}\left\|S_{i}^{2}\right)$		
Store: $U_{i}M_{1}$, $U_{i}M_{2}$, $U_{i}M_{3}$, $U_{i}M_{4}$, $U_{i}M_{5}$		
{Sensor Node Registration}		
	**(2)** Generate: $r_{GW}^{4}$	⟵**(1)** Transmit: $SID_{j}$
	Compute:	
	$TSID_{j}=h(SID_{j}\|r_{GW}^{4}\left\|K_{GW}\right)$	
	Store: $SID_{j}$, $TSID_{j}$	
	Delete: $S_{i}^{2}$	
	Transmit: $TSID_{j}$, $TID_{i}$, $S_{i}^{2}$ ⟶	**(3)** Store: $SID_{j}$, $TSID_{j}$, $TID_{i}$, $S_{i}^{2}$
{Authentication and Key distribution}		
**(1)** Verify Password		
Enter: $ID_{i}$, $PW_{i}$, $B_{i}$		
Compute: $R_{i}=Rep\left(B_{i},R_{bi}\right)$,	**(2)** Verify the user	
$r_{U}^{1}=h(PW_{i}\|R_{i}\left\|ID_{i}\right)⊕U_{i}M_{4}$,	Retrieve: $r_{U}^{2}=U_{i}M_{6}⊕S_{i}^{1}$	
$HPW_{i}=h(PW_{i}\|R_{i}\left\|r_{U}^{1}\right)$	Compute:	**(3)** Verify the gateway
$TID_{i}=U_{i}M_{1}⊕HPW_{i}$,	$U_{i}M_{8}^{*}=h(r_{U}^{2}\|TID_{i}\left\|ts_{U}\right)$	Retrieve: $r_{GW}^{5}=GM_{1}⊕TSID_{j}$
$S_{i}^{1}=U_{i}M_{2}⊕HPW_{i}$,	Verify: $r_{U}^{2}$’s freshness, $ts_{U}$	$TID_{i}=GM_{2}⊕r_{GW}^{5}$
$S_{i}^{2}=U_{i}M_{3}⊕HPW_{i}$,	$U_{i}M_{8}$= ?$U_{i}M_{8}^{*}$	$TID_{i}^{new}=GM_{3}⊕r_{GW}^{5}$
$U_{i}M_{5}^{*}=h(r_{U}^{1}\|TID_{i}\|S_{i}^{1}\left\|S_{i}^{2}\right)$		$r_{U}^{3}=U_{i}M_{7}⊕S_{i}^{2}$
Verify: $U_{i}M_{5}$ = ? $U_{i}M_{5}^{*}$	Generate: $r_{GW}^{5}$	Compute:
	$TID_{i}^{new}=h(ID_{i}\|r_{GW}^{5}\left\|K_{GW}\right)$,	$GM_{4}^{*}$
Generate: $r_{U}^{2}$, $r_{U}^{3}$, $ts_{U}$	$GM_{1}=TSID_{j}⊕r_{GW}^{5}$,	$=h(r_{GW}^{5}\|TSID_{j}\|TID_{i}\|TID_{i}^{new})$
Compute:	$GM_{2}=TID_{i}⊕r_{GW}^{5}$	Verify: $GM_{4}$= ?$GM_{4}^{*}$
$U_{i}M_{6}=r_{U}^{2}⊕S_{i}^{1}$,	$GM_{3}=TID_{i}^{new}⊕r_{GW}^{5}$	
$U_{i}M_{7}=r_{U}^{3}⊕S_{i}^{2}$	$GM_{4}$	
$U_{i}M_{8}=h(r_{U}^{2}\|TID_{i}\left\|ts_{U}\right)$	$=h(r_{GW}^{5}\|TSID_{j}\|TID_{i}\|TID_{i}^{new})$	
Transmit:	Transmit: $GM_{1}$, $GM_{2}$, $GM_{3}$	
$U_{i}M_{6}$, $U_{i}M_{7}$, $U_{i}M_{8}$, $TID_{i}$, $ts_{U}$⟶	$GM_{4}$, $U_{i}M_{7}$⟶	Generate: SK
		Compute:
	**(4)** Verify the sensor node	$SN_{j}M_{1}=SK⊕S_{i}^{2}$
**(5)** Verify the gateway	Compute:	$SN_{j}M_{2}=h(SK\|TID_{i}^{new}\left\|r_{U}^{3}\right)$
Compute:	$SN_{j}M_{3}^{*}$	$SN_{j}M_{3}$
$TID_{i}^{new}=GM_{6}⊕S_{i}^{1}$,	$=h(SN_{j}M_{1}\|SN_{j}M_{2}\|TSID_{j})$	$=h(SN_{j}M_{1}\|SN_{j}M_{2}\|TSID_{j})$
$GM_{5}^{*}=h\left(TID_{i}\|TID_{i}^{new}\right)$	Verify: $SN_{j}M_{3}$ = ? $SN_{j}M_{3}^{*}$	Replace: $TID_{i}\leftarrow TID_{i}^{new}$
$U_{i}M_{1}^{new}=TID_{i}^{new}⊕HPW_{i}$		Transmit: $SN_{j}M_{1}$, $SN_{j}M_{2}$
Verify: $GM_{5}$ = ? $GM_{5}^{*}$	Compute:	⟵$SN_{j}M_{3}$
Replace: $U_{i}M_{1}\leftarrow U_{i}M_{1}^{new}$	$GM_{5}=h\left(TID_{i}\|TID_{i}^{new}\right)$,	
	$GM_{6}=TID_{i}^{new}⊕S_{i}^{1}$	
Retrieve: $SK=SN_{j}M_{1}⊕S_{i}^{2}$	Replace: $TID_{i}\leftarrow TID_{i}^{new}$	
Compute:	Transmit: $GM_{5}$, $GM_{6}$, $SN_{j}M_{1}$,	
$SN_{j}M_{2}^{*}=h(SK\|TID_{i}^{new}\left\|r_{U}^{3}\right)$	⟵$SN_{j}M_{2}$	
Verify: $SN_{j}M_{2}$ = ? $SN_{j}M_{2}^{*}$		

**Table 5 sensors-23-01122-t005:** Definitions of channels, variables and other related parameters.

(*—-channels—-*)
free privateChannel1:channel [private].
free privateChannel2:channel [private].
free publicChannel1:channel.
free publicChannel2:channel.
(*—-constants—-*)
free Ri:bitstring [private].
free PWi:bitstring [private].
free IDi:bitstring [private].
free kgw:bitstring [private].
free IDg:bitstring.
free SIDj:bitstring.
(*—-shared key—-*)
free SK:bitstring [private].
(*—-functions—-*)
fun xor(bitstring, bitstring):bitstring.
fun concat(bitstring, bitstring):bitstring.
fun h(bitstring):bitstring.
(*—-events—-*)
event startUi(bitstring).
event endUi(bitstring).
event startGW(bitstring).
event endGW(bitstring).
event startSNj(bitstring).
event endSNj(bitstring).

**Table 6 sensors-23-01122-t006:** User’s process.

(*—-Ui process—-*)
let Ui =
new ru1:bitstring;
let HPWi = h(concat(concat(PWi, Ri), ru1)) in
out(privateChannel1,(IDi));
in(privateChannel1, (XTIDi:bitstring, XSi1:bitstring, XSi2: bitstring));
let UiM1= xor(XTIDi, HPWi) in
let UiM2= xor(XSi1, HPWi) in
let UiM3= xor(XSi2, HPWi) in
let UiM4 = xor(h(concat(concat(PWi, Ri), IDi)), ru1) in
let UiM5 = h(concat(concat(concat(ru1, XTIDi), XSi1), XSi2)) in
event startUi(IDi);
let ru1 = xor(h(concat(concat(PWi, Ri), IDi)), UiM4) in
let HPWi = h(concat(concat(PWi, Ri), ru1)) in
let XTIDi = xor(UiM1, HPWi) in
let XSi1 = xor(UiM2, HPWi) in
let XSi2 = xor(UiM3, HPWi) in
if h(concat(concat(concat(ru1, XTIDi), XSi1), XSi2)) = UiM5 then
new ru2:bitstring;
new ru3:bitstring;
new tsU:bitstring;
let UiM6 = xor(ru2, XSi1) in
let UiM7 = xor(ru3, XSi2) in
let UiM8 = h(concat(ru2, concat(XTIDi, tsU))) in
out(publicChannel1, (UiM6, UiM7, UiM8, XTIDi, tsU));
in(publicChannel1, (XGM5:bitstring, XGM6:bitstring, XXSNjM1:bitstring, XXSNjM2:bitstring));
let XTIDinew = xor(XGM6, XSi1) in
let UiM1new = xor(XTIDinew, HPWi) in
if h(concat(XTIDi, XTIDinew)) = XGM5 then
let UiM1 = UiM1new in
let SK = xor(XXSNjM1, XSi2) in
if(concat(concat(SK, XTIDinew), ru3)) = XXSNjM2 then
event endUi(IDi).

**Table 7 sensors-23-01122-t007:** Gateway’s process.

(*—-GW process—-*)
let GW =
in(privateChannel1, XIDi:bitstring);
new rgw1:bitstring;
new rgw2:bitstring;
new rgw3:bitstring;
let TIDi = h(concat(concat(IDi, rgw1), kgw)) in
let Si1 = h(concat(concat(IDi, rgw2), kgw)) in
let Si2 =h(concat(concat(IDi, rgw1), kgw)) in
out(privateChannel1, (TIDi, Si1, Si2));
in(privateChannel2, XSIDj:bitstring);
new rgw4:bitstring;
let TSIDj = h(concat(concat(XSIDj, rgw4), kgw)) in
out(privateChannel2, (TSIDj, TIDi, Si2));
event startGW(IDg);
in(publicChannel1, (XUiM6:bitstring, XUiM7:bitstring, XUiM8:bitstring, XTID:bitstring));
let Xru2 = xor(XUiM6, Si1) in
if h(concat(Xru2, TIDi)) = XUiM8 then
new rgw5:bitstring;
let TIDinew = h(concat(concat(XIDi, rgw5), kgw)) in
let GM1 = xor(TSIDj, rgw5) in
let GM2 = xor(TIDi, rgw5) in
let GM3 = xor(TIDinew, rgw5) in
let GM4 = h(concat(concat(concat(rgw5, TSIDj),TIDi), TIDinew)) in
out(publicChannel2, (GM1, GM2, GM3, GM4, XUiM7));
in(publicChannel2, (XSNjM1:bitstring, XSNjM2:bitstring, XSNjM3:bitstring));
if h(concat(concat(XSNjM1, XSNjM2), TSIDj)) =XSNjM3 then
let GM5 = h(concat(TIDi, TIDinew)) in
let GM6 = xor(TIDinew, Si1) in
let TIDi = TIDinew in
out(publicChannel1, (GM5, GM6, XSNjM1, XSNjM2));
event endGW(IDg).

**Table 8 sensors-23-01122-t008:** Sensor node’s process.

(*—-SNj process—-*)
let SNj =
out(privateChannel2, SIDj);
in(privateChannel2, (XTSIDj:bitstring, XXTIDi:bitstring, XXSi2:bitstring));
event startSNj(SIDj);
in(publicChannel2, (XGM1:bitstring, XGM2:bitstring, XGM3:bitstring, XGM4:bitstring,
XXUiM7:bitstring));
let Xrgw5 = xor(XGM1, XTSIDj) in
let XXTIDi = xor(XGM2, Xrgw5) in
let XXTIDinew = xor(XGM3, Xrgw5) in
let Xru3 = xor(XXUiM7, XXSi2) in
if h(concat(concat(concat(Xrgw5, XTSIDj), XXTIDi), XXTIDinew)) = XGM4 then
new SK:bitstring;
let SNjM1 = xor(SK, XXSi2) in
let SNjM2 = h(concat(concat(SK, XXTIDinew), Xru3)) in
let SNjM3 = h(concat(concat(SNjM1, SNjM2), XTSIDj)) in
let XXTIDi = XXTIDinew in
out(publicChannel2, (SNjM1, SNjM2, SNjM3));
event endSNj(SIDj).

**Table 9 sensors-23-01122-t009:** Queries and main process.

(*—-queries—-*)
query idi:bitstring; inj-event(endUi(idi)) ==> inj-event(startUi(idi)).
query idg:bitstring; inj-event(endGW(idg)) ==> inj-event(startGW(idg)).
query snj:bitstring; inj-event(endSNj(snj)) ==> inj-event(startSNj(snj)).
query attacker(SK).
(*—-process—-*)
process
((!Ui)|(!GW)|(!SNj))

**Table 10 sensors-23-01122-t010:** Result.

Verification summary:
Query inj-event(endUi(idi)) ==> inj-event(startUi(idi)) is true.
Query inj-event(endGW(idg)) ==> inj-event(startGW(idg)) is true.
Query inj-event(endSNj(snj)) ==> inj-event(startSNj(snj)) is true.
Query not attacker(SK[]) is true.

**Table 11 sensors-23-01122-t011:** Comparisons of the security features.

Security Features	Wu et al. [12]	Li et al. [15]	Masud et al. [6]	Ours
1. Resist Offline Password Guessing Attack	O	O	X	O
2. Resist Privileged Insider Attack	O	O	X	O
3. Resist User Impersonation Attack	X	O	O	O
4. Resist Server Impersonation Attack	O	X	O	O
5. Resist Replay Attack	X	X	X	O
6. Resist Man-in-the-Middle Attack	O	O	O	O
7. Resist Session Key Disclosure Attack	X	O	O	O
8. Preserve Forward Secrecy and Backward Secrecy	O	O	O	O
9. Provide Mutual Authentication	O	O	O	O
10. Preserve Anonymity	X	X	X	O

**Table 12 sensors-23-01122-t012:** Computational cost of cryptographic calculations (ms).

Symbol	Meaning	Time
$T_{m}$	The computational cost of multiplication in the field.	7.3529 [28]
$T_{bh}$	The computational cost of the biohash function operation.	7.3529 [29]
$T_{h}$	The computational cost of the one-way hash function operation.	0.0004 [28]

**Table 13 sensors-23-01122-t013:** Comparisons of computational cost.

Schemes	Wu et al. [12]	Li et al. [15]	Masud et al. [6]	Ours
User	$T_{bh}+10T_{h}+2T_{m}$	$T_{bh}+10T_{h}+2T_{m}$	$3T_{h}$	$T_{bh}+6T_{h}$
	=22.0627	=22.0627	=0.0012	=7.3553
GWN	$8T_{h}$	$8T_{h}+T_{m}$	$4T_{h}$	$5T_{h}$
	=0.0032	=7.3561	=0.0016	=0.002
Sensor Node	$2T_{h}+2T_{m}$	$4T_{h}+2T_{m}$	$2T_{h}$	$3T_{h}$
	=14.7066	=14.7074	=0.008	=0.0012
Total	$T_{bh}+20T_{h}+4T_{m}$	$T_{bh}+22T_{h}+5T_{m}$	$9T_{h}$	$T_{bh}+14T_{h}$
	=36.7725	=44.1262	=0.0036	=7.3585

## Data Availability

Not applicable.

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
