# Peer review of "An Improved Lightweight User Authentication Scheme for the Internet of Medical Things"

_sensors, 2023, doi:10.3390/s23031122_

Round 1

Reviewer 1 Report

Authors have really done commendable work based on IoT authentication scheme, but still there are a few issues to be addressed.

1.         The abstract indicates the clear condition & issues of lightweight user authentication model. But the objective is not clear and if it not given as separate subsection then it is difficult to recognize. So for better reading Objectives must be included. Better to be in the bulletin.

2.         Authors have claimed that they are trying to sort out the issues like, offline password guessing, impersonation, and privileged insider attacks from the article proposed by Masud et al; But in their article, they have clearly mentioned that their work only focuses on IoT several threats such as a denial of service attack, man-in-the-middle attack, and modification attack to the IoT networks’ security and privacy. This means that they haven't generalized the attacks. Then why have the authors pin pointed that they failed to address issues like offline password guessing, impersonation etc.? This questions the entire motive of the work. 

3.         For all the defined advanced method the authors tried to prove it with experimental scenarios. But evaluation metrics for the remaining scenarios need to be present. Therefore the outcome section can be extended. 

4.         The following healthcare security paper using IoT sensors paper can be referred.

https://doi.org/10.1155/2022/8457116 

5.         In the entire paper many grammatical mistakes are there. The paper should either be in active or passive voice, but not both.  The authors must check the quality of the writing.

6.         Conclusion section must be enhanced with separate sub-section like the pros and cons of this scheme. Why the author’s haven’t analysed the possible drawbacks of their own work?

7. How this work can be further enhanced in the future. There should be a few lines for it in future work.

Author Response

Original Manuscript ID: sensors-2167495      

Original Article Title: "An improved Lightweight User Authentication Scheme for the Internet of Medical Things"

Dear Editors and Reviewers,

Thank you for all your efforts in handling our manuscript “An improved Lightweight User Authentication Scheme for the Internet of Medical Things”. We carefully tried to address all comments raised by the reviewers.

This letter explains how we addressed those comments and questions. We feel that the manuscript has been significantly improved after our revision. We hope that the revised manuscript can now be considered acceptable for being published in Sensors.

The following pages explain point by point, the changes that we have made in response to the reviewer’s comments.

Your sincerely,

Jihyeon Ryu

Reviewer 2 Report

This manuscript presents the improved Lightweight User Authentication Scheme for Internet of Medical Things. The authors of this manuscript must focus on the following points in revised version.

1. Abstract shows that the proposed scheme performance is 266.49% better  than other studies, however, the result doesn't show this percentage. Please clarify this issue.

2. Introduction is not written properly. 

3. Related works should be a separate section. Compare the previous studies in a table.

4. The way to present the methodology is not clear. Use any standard methodology.

5. The data shows in table 11 and 12 is not enough to measure the performance.

6. How is this paper written specially for Internet of medical things. please specify.

Author Response

(The authors gave the same response as above.)

Reviewer 3 Report

The authors argue with an interesting and important topics related to the safety of Internet of Medical Thinks solutions. At the beginning of the article, the authors present a very short Introduction and also just as short Related Works. Both of these chapters should be extended. In particular, this remark concerns Related Works. The topic is not so unique, and you can certainly find from a dozen to several dozen works of other teams to refer to these works, indicating what the work of the authors of articles is better or different.

This is necessary despite the fact that they later refer to the Masuda scheme in detail.

The solutions proposed by the authors look interesting. Although they present security and performance analysis, it would certainly be necessary for a full picture in specific solutions for a full picture.

Unfortunately, the authors do not follow the structure of Imrad for the article at all. Therefore, the article does not have a secrete chapter discussion. It is true that its elements can be found in all subsections while reading the article, it makes the perception of the whole difficult.

However, the conclusions chapter itself looks like ... as if the authors ran out of strength and created a super laconic summary of the article. They should expand this chapter, emphasizing what they have achieved, to be achieved to profits from powerful applications and possible work for the future.

Author Response

(The authors gave the same response as above.)

Round 2

Reviewer 1 Report

The authors have justified my comments and hence the paper can be accepted in its current form.

Author Response

Dear Editors and Reviewers,

Thank you for all your efforts in handling our manuscript “An improved Lightweight User Authentication Scheme for the Internet of Medical Things”. We carefully tried to address all comments raised by the reviewers.

This letter explains how we addressed those comments and questions. We feel that the manuscript has been significantly improved after our revision. We hope that the revised manuscript can now be considered acceptable for being published in Sensors.

The following pages explain point by point, the changes that we have made in response to the reviewer’s comments.

Your sincerely,

Jihyeon Ryu

Reviewer 2 Report

The authors of this manuscript have revised their manuscript according to my previous comments. However, the authors are suggested to discuss their findings in a separate discussion section before the conclusion.

Author Response

(The authors gave the same response as above.)

Reviewer 3 Report

thank you, the authors answered my suggestions and questions.

Author Response

(The authors gave the same response as above.)
